# The proportion and determinants of COVID-19 infection among medical doctors in Sudan, 2020: A cross-sectional survey

**Maan Kabbashi**[1]*, **Amna Khairy**[2], **Amal Mohamed**[3], **Duha Abuobaida**[4], **Elfatih Malik**[5]

**1** Head of Quality Department, Central Ambulance Administration, General Directorate of Curative Medicine, Khartoum State Ministry of Health, Khartoum, Sudan, **2** Expanded Program of Immunization, Maternal and Child Health Directorate, Directorate General of Primary Health Care, Federal Ministry of Health, Khartoum, Sudan, **3** Head of Maternal Deaths Surveillance and Response Unit, National Reproductive Health Program, Mother and Child Health Directorate, General Directorate of Primary Health Care, Federal Ministry of Health, Khartoum, Sudan, **4** Malaria and NTDs Unit, Case Management Department, Diseases Control Directorate, Federal Ministry of Health, Khartoum, Sudan, **5** Faculty of Medicine Community Medicine Department, University of Khartoum, Khartoum, Sudan

* maankabbashi@gmail.com

## Abstract

Globally, frontline health care providers are among the most affected population group by the COVID-19 pandemic. Knowing the factors contributing to the transmission of COVID-19 infection among frontline health care providers is essential for implementing tailored control measures and protecting this vital population group. This study aimed to estimate the proportion and to identify factors associated with COVID-19 infection among medical doctors in Sudan. A web-based survey was used to collect data from medical doctors who were working in Sudan during the study period. Data were analyzed using SPSS® version 25; Descriptive analysis in terms of means (SD) for continuous variables, frequencies, and percentages with 95% CI for the categorical variable was conducted. Chi-square test and binary logistic regression for associations between the outcome variables (risk of exposure to COVID-19 infection and ever tested positive for COVID-19 infection) and independent variables (socio-demographic and infection control standards) were also performed. Out of 352 valid responses, 13.6% had tested positive for COVID-19 infection at least once during the pandemic. More than one-third have identified colleagues as the main sources of infection compared to 21% of patients (p-value < 0.04). Doctors who received training on COVID-19 were 60% less likely to have positive tests for COVID-19 (p-value <0.03), while lack of PPE and hand hygiene utilities had no statistically significant associations with testing positive for COVID-19 infection. In conclusion, a significant proportion of doctors have contracted COVID-19 infection from their colleagues. This calls for restricting infection control practices at hospitals, doctor's doormats, and any other shared places that allow day-to-day interaction between doctors and their colleagues. Also, urgent need for training doctors on COVID-19 infection control practices as it has been identified as the key protective factor.

**Data Availability Statement:** All relevant data are within the manuscript and its Supporting Information files.

**Funding:** The author(s) received no specific funding for this work

**Competing interests:** The authors have declared that no competing interests exist.

## Introduction

In December 2019, China reported a cluster of cases of pneumonia in Wuhan, Hubei Province. A novel coronavirus was eventually identified (SARS-CoV-2) and later named coronavirus disease 2019 (COVID-19) [1,2]. There is still a debate on how the virus started in the first place [3,4]. The virus causes symptoms that range from asymptomatic to severe infection and death. The COVID-19 pandemic has affected health systems across the globe, causing a reduction in health service delivery, vaccination, and an increase in mental health and domestic violence issues[5,6].

COVID-19 poses an occupational risk among essential workers with a higher incidence among health care providers [7–10]. In April 2020, the Center for Diseases Control and Prevention (CDC) Morbidity and Mortality Report documented that Health Care Workers (HCWs) made up to 3% of all COVID-19 cases in the United States [9]. The risk factors for COVID-19 transmission among HCWs are working in the high-risk department, longer duty hours with unprotected and prolonged patient contacts, lack of PPE, some aerosol-generating procedures, and suboptimal hand hygiene after contact with patients [11–13]. Most HCWs patients were not hospitalized; however, severe outcomes, including death, were reported among all age groups [14].

According to the World Health Organization Eastern Mediterranean Regional Office (WHO EMRO), it is estimated that the proportion of infection among healthcare workers with COVID-19 is up to 20% [15]; however, due to scarcity of data and absence of systematic reporting, this number may be underestimated. In Sudan, there are no official reports about the proportion/prevalence of COVID-19 among medical doctors or other medical staff. The spread of COVID-19 among HCWs will not only lead to increasing the number of COVID-19 cases but also to interruption in health care services delivery because the medical doctors who could provide these services are isolated or dead. This study aimed to estimate the proportion of suspected and confirmed medical doctors and to identify factors associated with transmission of COVID-19 among frontline health care providers in Sudan.

## Material and methods

This is a descriptive cross-sectional online survey conducted among Sudanese medical doctors working in Sudan in the period after the first COVID-19 case was detected in Sudan on March 12, 2020. In Sudan, according to the health emergency and epidemic control daily situation report, there are 28 COVID-19 isolation centers; nine of them are in Khartoum (Unpublished data).

### Study questionnaire and data collection

Data were collected during the period from 24th June to 27th October 2020. The questionnaire was designed in Google Forms, and the link was distributed through online platforms targeting medical doctors working at hospitals in Sudan during the pandemic. The link was re-shared daily in 10 Facebook doctor's groups/pages. The average group's members ranged from 500 to 100000. Also, the What's App and telegram groups for 10 medical associations and Sudan Medical Specialization Board (SMSB) registrars were targeted, with average members from 100–400 per group. Personal contact for a list of members was made and two reminders were sent for each contact.

### Study variables

**The outcome variables.** Two outcome variables: Exposure risk to COVID-19 infection [16] (High or Low) and COVID-19 status (ever tested positive for SARs-COV-2 virus).

Regarding the exposure risk to COVID-19 infection; medical doctors were classified as "high-risk exposure to COVID-19 infection" if they answered yes to any of the following variables: ever been in contact with a suspected or confirmed COVID-19 case/s within 2 meters for more than half-hour without a face mask, ever been in physical contact (without glove) with a suspected or confirmed COVID-19 case/s, ever been in physical contact with secretions or excretion of a suspected or confirmed COVID-19 case/s, and home visit for patients with respiratory symptoms. The definition of the suspected COVID-19 case was adopted from the National COVID-19 case management guidelines. The latter considered patients with clinical scores 5 or more as a suspected case for COVID-19, and all Infection Prevention and Control (IPC) measures should be taken when treating this patient [17].

The COVID-19 status was measured as a dichotomous variable: ever tested positive for SARs-COV-2 virus before? Yes, No. This includes both previous and current COVID-19 infections.

**Independent variables.** Include socio-demographic, source of infection, post-contact quarantine status, training related to COVID-19, and hospital type. Variables related to IPC standards at the hospital: availability of triage and isolation room, availability of PPE, availability of water, sanitizers for hand hygiene, and availability of Federal Ministry of Health (FMOH) COVID-19 guidelines.

## Data analysis

Data was downloaded as an excel sheet and cleaned, SPSS® version 25 was used for the analysis. Descriptive analysis in terms of means (SD) for continuous variables and frequencies and percentages with 95% confidence interval (CI) for a categorical variable were conducted. Bivariate analysis using Pearson's Chi-square test was done to test the associations between the two outcome variables (exposure risk to COVID-19 and COVID-19 status) with the independent variables. These associations have been further adjusted using multivariable binary logistic regression analysis. Adjusted Odds Ratio (OR) and 95% CI for the associations between outcome variables and independent variables were obtained.

## Ethical considerations

Written informed consent was obtained from the participants. This was obtained by the statement that describes the research purpose and the voluntariness of participation in the survey. Contact for further queries about the research was included at the end of the informed consent statement. Ethical approval to conduct the study was obtained from the National Ethics Committee, FMOH Sudan.

## Results

The total number of valid responses was 352. The mean (SD) age of responders was 29.6 (± 4.4). Fifty percent of the population were males, 58.5% (n = 206) were working in public health facilities and 89.5% (n = 317) were working in Khartoum State -the capital of Sudan-. The main site of work was Emergency Room (ER) and general ward (44% (n = 155) and 26.1% (n = 92) respectively). Almost half of the respondents (n = 175) received some sort of training related to COVID-19 (Table 1).

Forty-five percent (n = 159) of the participants had been suspected of COVID-19. Those isolated were 43.8% (n = 154). While less than one-third of the respondents (n = 93) have been tested for COVID-19 using Reverse Transcription—Polymerase Chain Reaction (RT-PCR), 51.6% (n = 48) of them were tested positive (Table 2). The source of infection was mainly from

**Table 1. Demographic characteristics of the participants (n = 352).**

| Variable | Frequency (%) |
|---|---|
| Age Mean± SD | 29.63 ± 4.45 |
| Gender | |
| Male | 180 (51.1%) |
| Female | 172 (48.9%) |
| Location | |
| Khartoum state | 315 (89.5%) |
| Other states | 37 (10.5%) |
| Profession | |
| Specialists / Consultants | 12 (3.4%) |
| Registrars | 188 (53.4%) |
| Medical officers | 126 (35.8) |
| House officers | 26 (7.4%) |
| Working area | |
| ER | 155 (44%) |
| ICU/HDU/CCU | 43 (12.2%) |
| Wards | 92 (26.1%) |
| IPC | 1 (0.3%) |
| Administration | 19 (5.4%) |
| Surgery related | 16 (4.5%) |
| Other | 26 (7.4%) |
| Facility type | |
| Public | 206 (58.5%) |
| Private/ NGO | 103 (29.3%) |
| Military | 34 (9.7%) |
| Other | 9 (2.6%) |
| Taken any training related to COVID-19 | |
| Yes | 175 (49.7%) |

colleagues in 37.5% (n = 18) of respondents, followed by patients (20.8%, n = 10). (P<0.04) (Fig 1).

## Factors associated with exposure risk to COVID-19

The study showed a statistically significant gender difference (74% (n = 134) of males had higher exposure risk to COVID-19 infection compared to 64% (n = 110) of females) in risk of exposure to COVID-19 infection; OR = 0.5, 95% CI 0.4–0.9), p<0.02. On the contrary, Exposure risk was not affected by age (OR = 0.9, 95% CI 0.9–1.0), p> 0.55). Doctors working in ER had higher exposure to the infection compared to other work settings (p <0.03) (Table 3). Similarly, almost three-quarters of doctors who received training on COVID-19 had a high

**Table 2. Proportion of doctors, who have been suspected, isolated, tested, or have positive results and the source of infection.**

| Variable | Frequency (%) |
|---|---|
| Medical doctors suspected to have COVID-19 | 159 (45.2%) |
| Medical doctors isolated after suspicion for COVID-19 | 154 (43.8%) |
| Medical doctors being tested for COVID-19 | 93 (26.4%) |
| Medical doctors being tested positive for COVID-19 | 48 (13.6%) |

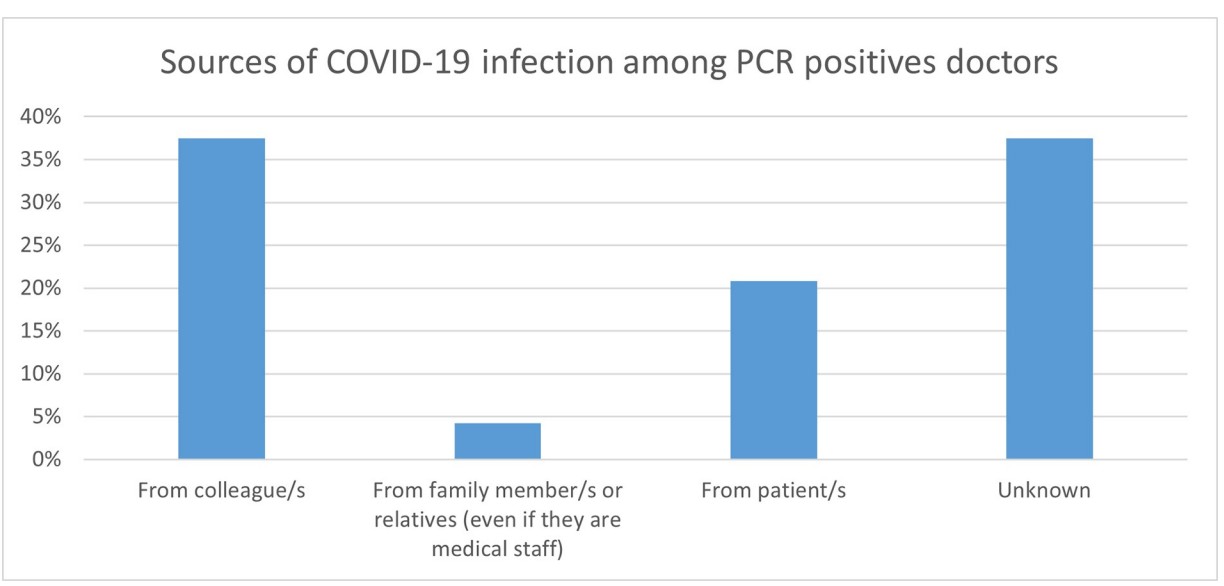

**Fig 1. Source of COVID-19 infection among the doctors who had ever tested positive for COVID-19 infection before.**

exposure risk (p<0.03) (Table 4). Also, exposure risk was high among respondents who had previously contracted COVID-19 infection from colleague/s and patient/s 18 (75.0%), 10 (76.9%), respectively. By contrast, it was not associated with hospital type (p > 0.11), availability of IPC measures or lack of PPE and hand hygiene utilities. p>0.053), p> 0.80), p>0.68) respectively (Table 3).

### Factors associated with confirmed COVID-19 infection among doctors

Doctors who received training on COVID-19 (either provided by the facility or not) were less likely to have a positive test for COVD-19. OR = 0.4(95% CI: 0.2–0.9), p<0.03 (Table 3). No statistically significant associations between confirmed COVID-19 infection with age or gender, p>0.38 and p>0.44 respectively. The proportion of confirmed COVID-19 infection among doctors who had high exposure risk was double that of doctors who had low-exposure risk, 16% (n = 39) and 8.3% (n = 9) respectively. However, the difference was not statistically significant (P> 0.05) (Table 5).

### Discussion

COVID- 19 diseases can be categorized as an occupational disease and appear to have been associated with significant mortality among doctors and healthcare workers globally [7,15,17,18]. When COVID-19 hits Sudan, the health care system was already challenged by political, economic, and other public health threats for its resilience [19]. We expected the risk factors for COVID-19 infection in Sudan to be unique for the context of the health care system in the country. Risk factors for COVID-19 among Health care workers/ doctors in other settings, might not apply to the health care system in Sudan.

Thus, it is important to understand the proportion and factors that influence the spread of disease among medical doctors in Sudan [19]. In this study, we studied COVID-19 infection among doctors in Sudan. Also, we identified key determinants of contracting COVID-19 infection among doctors in Sudan and females represented half of the study population. The majority of the study population was working in the emergency department and general ward.

**Table 3. Factors associated with risk of exposure to COVID-19.**

| Demographic and professional characteristics | Exposure to COVID | | X2 | P |
|---|---|---|---|---|
| | Low risk 108(30.7%) | High risk 244 (69.3%) | | |
| Gender | | | | |
| Male | 46 (25.6%) | 134 (74.4%) | 4.5 | 0.03 |
| Female | 62 (36.0%) | 110 (64.0%) | | |
| Work area | | | | |
| Emergency room | 44 (28.8%) | 109 (71.2%) | 9.8 | 0.03 |
| Critical care units (ICU, HDU, NICU, PICU) | 6 (14.6%) | 35 (85.4%) | | |
| Wards | 28 (30.8%) | 63 (69.2%) | | |
| Infection prevention and control | 0 (0%) | 1 (100%) | | |
| Hospital administration | 10 (52.6%) | 9 (47.4%) | | |
| Hospital Type | | | | |
| Public hospital | 57 (27.7%) | 149 (72.3%) | 2.1 | 0.16 |
| Non-Public | 51 (34.9%) | 95 (65.1%) | | |
| Training by self | | | | |
| No | 64 (36.2%) | 113 (63.8%) | 5 | 0.02 |
| Yes | 44 (25.1%) | 131 (74.9%) | | |
| Mean Age (SD) | 29.8 (4.1) | 29.5 (4.5) | 0.4* | 0.619 |
| PPE | | | | |
| No | 70 (84.3%) | 13 (15.7%) | 0.3 | 0.53 |
| Yes | 234 (87.0%) | 35 (13.0%) | | |
| Hand hygiene | | | | |
| No | 96 (85.7%) | 16 (14.3%) | 0 | 0.8 |
| Yes | 208 (86.7%) | 32 (13.3%) | | |
| Triage | | | | |
| I work in a designated hospital for COVID-19 | 33 (82.5%) | 7 (17.5%) | 0.7 | 0.68 |
| No | 82 (88.2%) | 11 (11.8%) | | |
| Yes | 189 (86.3%) | 30 (13.7%) | | |
| Temporal isolation | | | | |
| I work in a designated hospital for COVID-19 | 26 (81.3%) | 6 (18.8%) | 4.8 | 0.07 |
| No | 103 (92.0%) | 9 (8.0%) | | |
| Yes | 175 (84.1%) | 33 (15.9%) | | |
| Source of infection for positive | | | | |
| From colleague/s | 6 (25.0%) | 18 (75.0%) | 247.5 | 0 |
| From family member/s or relatives (even if they are medical staff) | 2 (50.0%) | 2 (50.05) | | |
| From patient/s | 3 (23.1%) | 10 (76.9%) | | |
| I do not know exactly | 6 (25.0%) | 18(75.0%) | | |
| Not tested/ Negative PCR | 287 (100.0%) | 0 (0.0%) | | |

Exposure risk to COVID-19 infection was 50 percent higher among male doctors. This finding could be related to Sudanese studies that found females were more knowledgeable and had more positive attitudes towards COVID-19 [20,21].

The proportion of isolated staff was 43.8% of the participant. The decision of the isolation was made after risk assessment in 9.7% of cases; this may conclude that some of the self-isolated or isolated doctors were unnecessarily isolated.

One-third of respondents were suspected as COVID-19 case with 14% proportion of positively tested respondents, this proportion is going with the statement by the WHO EMRO

**Table 4. Shows a binary logistic regression between high risks of exposure to COVID-19 and the positive result for the SARS-CoV-2and the independent variables.**

|  | High risk of exposure to COVID-19 | | A positive result of COVID-19 | |
|---|---|---|---|---|
|  | OR (95% CI) | P | OR (95%CI) | P |
| Gender (Male) | 0.5 (0.4–0.9) | 0.02 | 0.7(0.4–1.4) | 0.38 |
| Age | 0.9 (0.9–1.0) | 0.55 | 0.9(0.8–1.0) | 0.50 |
| Career status | 1.0 (0.7–1.0) | 0.56 | 1(0.6–1.9) | 0.63 |
| Hospital type | 0.7 (0.4–1.0) | 0.11 | 1.0(0.5–2.0) | 0.76 |
| Training | 0.5 (0.4–0.9) | 0.02 | 0.4(0.2–0.9) | 0.03 |

region director[15]. In Europe, health care workers have accounted for a substantial proportion of all COVID-19 cases. In Italy, health care workers accounted for up to 10% of cases. while in Spain it was estimated to be 26% [22].

This study found that colleagues were the major source of contracting the infection followed by contact with patients. This finding was contradicting reports that found patients to be the main source of COVID-19 infection among health care workers [11,12,15]. However, it also goes with the study that reported health care worker contact with COVID-19 patients not in health care settings particularly household and community settings [23]. While in Rome study stated that nearly half of positive COVID-19 health care workers did not report previous exposure to infected COVID-19 subjects. [24]. Considering health care workers as a source of infection [24,25], further protection is important to minimize transmitting the infection to patients and colleagues. Most of those who tested positive in this study were working in the emergency room, this can be explained by the stress and long duties compared to working in the wards that leave them less likely to commit to infection control measures [25]. Two studies found that the incidence of infection was higher inwards and health followed by emergency department and critical care [22,25]. This study has not found any significant change in prevalence based on the workplace; however, it was found that there is a higher risk of exposure to the disease when working in critical care units, followed by emergency rooms and wards.

This study has found that those who received training related to COVID-19, were 60% less likely to have a positive test for COVD-19, this may reflect the importance of training in COVID-19 among other preventive measures. Surprisingly, this study has not found any association between getting COVID-19 infection and availability of IPC, PPEs, hand hygiene kits, triage, and temporal isolation availability in the hospital. This finding is contrary to what was found in Wuhan and California [11,12]. Proper use of protective measures has dramatically reduced infections among health workers [22] that and is associated with shorter duration and less severe illness [26]. But, this finding can be explained by the fact that PPE is not effective unless used correctly while dressing and undressing [25,27].

## Conclusion

Novel coronavirus infection is considered an occupational disease to health care workers. Understanding the prevalence and risk factors associated with the infection will assist in the protection of frontline workers. This study has found that COVID-19 was more prevalent among male doctors, who contracted the infection from colleagues. Moreover, that training about COVID-19 acts as a protective factor. Protective interventions including vaccines should be provided to healthcare workers to prevent the spread of disease, reduce mortality and maintain the health system. Further studies are needed to identify the incidence of COVID-19 among Sudanese doctors after the vaccine and to understand the relationship between COVID-19 and infection control measures availability in Sudanese hospitals.

**Table 5. Factors associated with confirmed COVID-19 infection among doctors.**

| Variables | Test results | | X2 | P |
|---|---|---|---|---|
| | **Negative results 304 (86.4%)** | **Positive results 9 (8.3%)** | | |
| Risk category | | | | |
| Low risk | 99 (91.7%) | 9 (8.3%) | 3.6 | 0.05 |
| High risk | 205 (84.0%) | 39 (16.0%) | | |
| Gender | | | | |
| Male | 153 (85.0%) | 27 (15.0%) | 0.5 | 0.44 |
| Female | 151 (87.8%) | 21 (12.2%) | | |
| Work area | | | | |
| Emergency room | 131 (85.6%) | 22 (14.4%) | 1.7 | 0.77 |
| Critical care units | 35 (85.4%) | 6 (14.6%) | | |
| Wards | 76 (83.5%) | 15 (16.5%) | | |
| Infection prevention and control | 1 (100%) | 0 (0%) | | |
| Administration | 18 (94.7%) | 1 (5.3%) | | |
| Hospital type | | | | |
| Public hospital | 179 (86.9%) | 27 (13.1%) | 0.1 | 0.75 |
| Non-Public | 125 (85.6%) | 21 (14.4%) | | |
| Training by hospitals | | | | |
| No | 95 (90.5%) | 10 (9.5%) | 10.1 | 0 |
| Yes | 157 (81.3%) | 36 (18.7%) | | |
| Self-training (e.g. online course) | | | | |
| No | 160 (90.4%) | 17 (9.6%) | 4.9 | 0.03 |
| Yes | 144 (82.3%) | 31 (17.7%) | | |
| Age Mean (SD) | 29.7 (4.3) | 29.1 (5) | 0.8 | 0.38 |
| PPE | | | | |
| No | 28 (33.7%) | 55 (66.3%) | 0.4 | 0.49 |
| Yes | 80 (29.7%) | 189 (70.3%) | | |
| Hand hygiene | | | | |
| No | 30 (26.8%) | 82 (73.2%) | 1.1 | 0.27 |
| Yes | 78 (32.5%) | 162 (67.5%) | | |
| Triage | | | | |
| I work in a designated hospital for COVID-19 | 8 (20%) | 32 (80%) | 3.6 | 0.15 |
| No | 34 (36.6%) | 59 (63.4%) | | |
| Yes | 66 (30.1%) | 153 (69.9%) | | |
| Temporal isolation | | | | |
| I work in a designated hospital for COVID-19 | 6 (18.8%) | 26 (81.3%) | 4.1 | 0.12 |
| No | 41 (36.6%) | 71 (63.4%) | | |
| Yes | 61 (29.3%) | 147 (70.7%) | | |
| Source of infection for positive | | | | |
| From colleague/s | 2 (8.3%) | 22 (91.7%) | 9.2 | 0.04 |
| From family member/s or relatives (even if they are medical staff) | 0 (0%) | 4 (100%) | | |
| From patient/s | 3 (23.1%) | 10 (76.9%) | | |
| I do not know exactly | 6 (25% | 18 (75%) | | |
| Not tested/ Negative PCR | 97 (33.8%) | 190 (66.2%) | | |

### Limitations

Although internet access/use was recognized as a limiting factor to contribution in this survey, researchers have exerted much effort to get a representative sample. Still, the possibility of selection bias in online surveys cannot be ignored.

The survey showed not much risk difference for COVID-19 among those who Use PPE vs. those who do not. This might point out poor hand hygiene and PPE practices among doctors. However, the survey assessed hand hygiene and PPE practices among doctors only subjectively. A more objective assessment for PPE use and hand hygiene practice would minimize the potential misinformation bias of the subjective assessment.

## Supporting information

**S1 Table. IPC measures at hospitals.**
(DOCX)

**S1 Questionnaire. COVID-19 infection among medical doctors in Sudan.**
(DOCX)

**S1 Data. COVID-19 among doctors Sudan 2020.**
(SAV)

## Acknowledgments

To Sudanese doctors, the front-liners who are working under tough conditions, risking their health to save other's life.

## Author Contributions

**Conceptualization:** Maan Kabbashi, Amna Khairy, Amal Mohamed, Elfatih Malik.

**Formal analysis:** Amna Khairy, Amal Mohamed.

**Methodology:** Maan Kabbashi, Amna Khairy, Amal Mohamed, Duha Abuobaida, Elfatih Malik.

**Supervision:** Maan Kabbashi, Elfatih Malik.

**Validation:** Duha Abuobaida.

**Visualization:** Maan Kabbashi, Amna Khairy, Amal Mohamed, Elfatih Malik.

**Writing – original draft:** Maan Kabbashi, Amna Khairy, Amal Mohamed, Duha Abuobaida.

**Writing – review & editing:** Maan Kabbashi, Amna Khairy, Amal Mohamed, Duha Abuobaida.

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
