## [Decision Letter · Decision Letter 0]

9 Nov 2021

PONE-D-21-29938The Proportion and Determinants of COVID-19 Infection among Medical Doctors in Sudan, 2020: A Cross Sectional SurveyPLOS ONE

Dear Dr. Kabbashi,

Thank you for submitting your manuscript to PLOS ONE. After careful consideration, we feel that it has merit but does not fully meet PLOS ONE’s publication criteria as it currently stands. Therefore, we invite you to submit a revised version of the manuscript that addresses the points raised during the review process.

We look forward to receiving your revised manuscript.

Kind regards,

Sanjay Kumar Singh Patel, Ph.D.

Academic Editor

PLOS ONE

4. Please include additional information regarding the survey or questionnaire used in the study and ensure that you have provided sufficient details that others could replicate the analyses. For instance, if you developed a questionnaire as part of this study and it is not under a copyright more restrictive than CC-BY, please include a copy, in both the original language and English, as Supporting Information.

5. We suggest you thoroughly copyedit your manuscript for language usage, spelling, and grammar. If you do not know anyone who can help you do this, you may wish to consider employing a professional scientific editing service.

Whilst you may use any professional scientific editing service of your choice, PLOS has partnered with both American Journal Experts (AJE) and Editage to provide discounted services to PLOS authors. Both organizations have experience helping authors meet PLOS guidelines and can provide language editing, translation, manuscript formatting, and figure formatting to ensure your manuscript meets our submission guidelines. To take advantage of our partnership with AJE, visit the AJE website (http://aje.com/go/plos) for a 15% discount off AJE services. To take advantage of our partnership with Editage, visit the Editage website (www.editage.com) and enter referral code PLOSEDIT for a 15% discount off Editage services.  If the PLOS editorial team finds any language issues in text that either AJE or Editage has edited, the service provider will re-edit the text for free.

Reviewers' comments:

Reviewer's Responses to Questions

**Comments to the Author**

1. Is the manuscript technically sound, and do the data support the conclusions?

Reviewer #1: Yes

Reviewer #2: Yes

2. Has the statistical analysis been performed appropriately and rigorously? 

Reviewer #1: Yes

Reviewer #2: Yes

3. Have the authors made all data underlying the findings in their manuscript fully available?

Reviewer #1: Yes

Reviewer #2: Yes

4. Is the manuscript presented in an intelligible fashion and written in standard English?

Reviewer #1: No

Reviewer #2: Yes

5. Review Comments to the Author

Reviewer #1: The manuscript needs major changes to improve it further.

Comments:

1. The paper has large number of grammatical and English language errors that needs to be corrected in order to improve the manuscript. Many extra spaces and punctuation need to be taken care by the author.

2. Line 107 state that males have higher exposure risk than females, is it because there are more male HCWs or more male participated in the survey? Reason for the finding should be stated.

3. In table 4, not much risk difference is noticed among the ones who maintained hand hygienics also for the ones who wear PPE kit and those who doesn’t? Any justification and what is the error percentage of the survey results? Any biases associated with results could be highlighted.

4. Line 168, “those who received training related to COVID-19, were 60% less likely to have a positive test for COVD-19” contradict with the findings in the table 5. Justify?

5. What is X2 in the table 4 and 5 and its significance is not given?

6. Also, the representation of information in the form of tables could be improved by adding few graphs or pie charts to make it more explicit.

7. Why the subject of the study is chosen to be the Sudanese doctors and does the results are specific to Sudan or it could be beneficial to other HCWs all over the world. The author could discuss about it.

Reviewer #2: The manuscript by Kabbashi et al., “The Proportion and Determinants of COVID-19 Infection among Medical Doctors in Sudan, 2020: A Cross Sectional Survey” analyzed/surveyed to estimate the proportion and to identify factors associated with COVID-19 infection among medical doctors in Sudan. Data was analyzed using SPSS® version 25; Descriptive analysis in terms of means (SD) for continuous variables, frequencies and percentages with 95% CI for categorical variable was conducted. They found significant proportion of doctors have contracted COVID-19 infection from their colleagues. This calls for restrict infection control practices at hospitals, doctors’ doormats and any other shared places that allows day-to day interaction between doctors and their colleagues. This is an interesting study but it requires revision to address few minor concerns.

Recommendation: Publish after minor revisions.

Comments:

1. Whilst the content is good, the standard of English needs to be improved throughout the manuscript. This would enhance the quality of the manuscript.

2. The authors may provide at least one Figure as summary or prospect of this study.

3. Authors should add a paragraph to discuss limitations of this study.

---

## [Author Response · Author response to Decision Letter 0]

19 Apr 2022

Dear PLOS ONE editors and reviewers,

Thank you for your feedback. Below are points (in blue) addressing the author’s responses to editorial and reviewer’s comments: 

Response to Editorial Comments:

Editorial Comment (1): Please ensure that you include a title page within your main document: 

Authors’ Response: 

• The manuscript has been amended and the title page is included.

Editorial Comment (2): Please provide additional details regarding participant consent. In the ethics statement in the Methods and online submission information, please ensure that you have specified (1) whether consent was informed and (2) what type you obtained (for instance, written or verbal, and

if verbal, how it was documented and witnessed). If your study included minors, state whether you obtained consent from parents or guardians. If the need for consent was waived by the ethics committee,

please include this information.

 Authors’ response: 

• An ethical Consideration section was added at the end of the methodology section. “Ethical Consideration: Written informed consent was obtained from the participants. This was obtained by a statement that describes the research purpose and the Voluntariness of participation in the survey. Contact for further Queries about the research was included at the end of the informed consent statement. Ethical approval to conduct the study was obtained from the national ethics Committee, Federal Ministry of Health, Sudan. Also, the Ethical Approval letter was uploaded as supporting information.

Editorial Comment (3): Please include additional information regarding the survey or the questionnaire used in the study and ensure that you have provided sufficient details that others could replicate the analyses. For instance, if you developed a questionnaire as part of this study and it is not under copyright more restrictive than CC-BY, please include a copy, in both the original language and English, as Supporting Information.

 Author’s Response: 

• The subtitle in the methodology section: “data collection” was amended to “study Questionnaire and data collection”.

• The following statement was added “. A pretested pre-coded self-administered questionnaire developed for this study. It consisted of 40 Questions,”.

• A copy of the Questionnaire is attached as supplementary information. A caption for that file was included in the Manuscript file.

Editorial Comment (4): We suggest you thoroughly copyedit your manuscript for language

usage, spelling, and grammar. If you do not know anyone who can help you do this, you may wish to consider employing a professional scientific editing service. The name of the colleague or the details of the professional service that edited your manuscript. 

Author’s Response: 

• The paper has been subjected to English language proofreading by all authors and also by using Grammarly application to check spelling and crorrect grammar mistakes

Editorial Comment (5): In your Data Availability statement, you have not specified where the minimal data set underlying the results described in your manuscript can be found

Author’s Response: 

• I have uploaded study underlying data set as Supporting Information file

Editorial Comment (6): Please include captions for your Supporting Information files at the end of your manuscript, and update any in-text citations to match accordingly.

Author’s Response: 

• Captions for Supporting Information files are uploaded at the end of manuscript,

First Reviewer’s Comments: 

(1) The paper has a large number of grammatical and English language errors that need to be corrected to improve the manuscript. Many extra spaces and punctuation need to be taken care of by the author.

Author’s Response: 

• The paper has been subjected to English language proofreading by all authors and also by using Grammarly application to check spelling and crorrect grammar mistakes

 (2) Line 107 states that males have a higher exposure risk than females, is it because there are more male HCWs or more males participating in the survey? The reason for the finding should be stated.

Author’s Response: 

• Males to female ratio in the study = 180/172= almost 1. Hence the finding is unlikely to be due to disproportionate gender representation in the sample. However, in this survey, most of the doctors working in high-risk settings were males. A supplementary table for bivariate analysis for gender versus work experience has been added.

(3) In table 4, not much risk difference is noticed among the ones who maintained hand hygienic also for the ones who wear PPE kit and those who doesn’t? Any justification and what is the error percentage of the survey results? Any biases associated with results could be highlighted.

Author’s Response:

• The survey showed not much risk difference for COVID-19 among those who Use PPE vs. those who do not. This might point out poor hand hygiene and PPE practices among doctors. However, the survey assessed hand hygiene and PPE practices among doctors only subjectively. A more objective assessment for PPE use and hand hygiene practice would minimize the potential misinformation bias of the subjective assessment.

• The above paragraph has been added to the limitations.

4. Line 168, “those who received training related to COVID-19, were 60% less likely to have a positive test for COVD-19” contradict with the findings are in table 5. Justify?

• Both table (3) and table (5) show the association between training on COVID-19 and ever been tested positive for SARs-COV-2 infection. The findings in Table (5) are findings from the bivariate analysis. It shows that the majority of those who had any type of training related to COVID-19 infection never tested positive for SARs-COV-2 infection before (90.3% and 81.5%). However, in table (3) adjustment for this association for potential confounders was done using multiple logistic regression analyses. An adjusted odds ratio of 0.4 95%CI (0.2-0.9), P=0.03, represents the measurement of the strength, direction, and significance of these associations. The latter has been interpreted as those who received training related to COVID-19, were 60% less likely to have a positive test for COVD-19”

5. What is X2 in table 4 and 5 and its significance is not given?

Author’s Response: 

• X2 means Pearson’s chi Square, it is the outcome from the bivariate analysis.

• The explanation for this point has been added to the Methodology section.

6. Also, the representation of information in the form of tables could be improved by adding a few graphs or pie charts to make it more explicit.

Author’s Response: 

• A bar chart for one of the key findings: (Figure I: Source of COVID-19 infection among the doctors who tested positive for COVID-19 infection before) was added.

7. Why the subject of the study is chosen to be the Sudanese doctors and does the results are specific to Sudan or it could be beneficial to other HCWs all over the world. The author could discuss it.

Author’s Response: 

• When COVID-19 hit Sudan, the health care system was already challenged by political, economic, and other public health threats for its resilience. We expected the risk factors for COVID-19 infection in Sudan to be unique for the context of the health care system in the country. Risk factors for COVID-19 among Health care workers/ doctors in other settings, might not apply to the health care system in Sudan. 

• The above paragraph was added to the discussion section. 

Second reviewer’s Comments:

Reviewer #2: The manuscript by Kabbashi et al., “The Proportion and Determinants of COVID-19 Infection among Medical Doctors in Sudan, 2020: A Cross-Sectional Survey” analyzed/surveyed to estimate the proportion and to identify factors associated with COVID-19 infection among medical doctors in Sudan. Data were analyzed using the SPSS® version 25; Descriptive analysis in terms of means (SD) for continuous

variables, frequencies, and percentages with 95% CI for categorical variables was conducted. They found a significant proportion of doctors have contracted COVID-19 infection from their colleagues. This calls for restrict infection control practices at hospitals, doctors’ doormats and any other shared places that allow day-to-day

interaction between doctors and their colleagues. This is an interesting study but it requires revision to address a few minor concerns.

Recommendation: Publish after minor revisions.

(1)Whilst the content is good, the standard of English needs to be improved throughout the manuscript. This would enhance the quality of the manuscript.

Author’s Response: 

• The paper has been subjected to English language proofreading by all authors and also by using Grammarly application to check spelling and crorrect grammar mistakes

 (2) The authors may provide at least one Figure as a summary or prospect of this study.

Author’s Response: 

• A bar chart for one of the key findings: (Figure I: Source of COVID-19 infection among the doctors who tested positive for COVID-19 infection before) was added.

(3) Authors should add a paragraph to discuss the limitations of this study.

The section below for the limitations of the study was added to the manuscript: 

“Limitations: Although internet access/use was recognized as a limiting factor to contribution in this survey, researchers have exerted much effort to get a representative sample. Still, the possibility of selection bias in online surveys cannot be ignored by Internet access. The survey showed not much risk difference for COVID-19 among those who Use PPE vs. those who do not. This might point out poor hand hygiene and PPE practices among doctors. However, the survey assessed hand hygiene and PPE practices among doctors only subjectively. A more objective assessment for PPE use and hand hygiene practice would minimize the potential misinformation bias of the subjective assessment”.

---

## [Editor Report · Decision Letter 1]

21 Apr 2022

The Proportion and Determinants of COVID-19 Infection among Medical Doctors in Sudan, 2020: A Cross Sectional Survey

PONE-D-21-29938R1

Dear Dr. Kabbashi,

We’re pleased to inform you that your manuscript has been judged scientifically suitable for publication and will be formally accepted for publication once it meets all outstanding technical requirements.

Kind regards,

Sanjay Kumar Singh Patel, Ph.D.

Academic Editor

PLOS ONE

---

## [Editor Report · Acceptance letter]

28 Oct 2022

PONE-D-21-29938R1 

The Proportion and Determinants of COVID-19 Infection among Medical Doctors in Sudan, 2020: A Cross-Sectional Survey  

Dear Dr. Kabbashi:

I'm pleased to inform you that your manuscript has been deemed suitable for publication in PLOS ONE. Congratulations! Your manuscript is now with our production department. 

Kind regards, 

on behalf of

Dr. Sanjay Kumar Singh Patel 

%CORR_ED_EDITOR_ROLE%

PLOS ONE